# Duvernoy’s Gland Transcriptomics of the Plains Black-Headed Snake, *Tantilla nigriceps* (Squamata, Colubridae): Unearthing the Venom of Small Rear-Fanged Snakes

**DOI:** 10.3390/toxins13050336

**Published:** 2021-05-06

**Authors:** Erich P. Hofmann, Rhett M. Rautsaw, Andrew J. Mason, Jason L. Strickland, Christopher L. Parkinson

**Affiliations:** 1Department of Biological Sciences, Clemson University, Clemson, SC 29634, USA; ephofmann564@cfcc.edu (E.P.H.); rrautsa@clemson.edu (R.M.R.); mason.501@osu.edu (A.J.M.); jasonstrickland@southalabama.edu (J.L.S.); 2Department of Forestry and Environmental Conservation, Clemson University, Clemson, SC 29634, USA

**Keywords:** colubrid, Duvernoy’s gland secretion, rear-fanged snake, RNA-seq, transcriptomics, venom

## Abstract

The venoms of small rear-fanged snakes (RFS) remain largely unexplored, despite increased recognition of their importance in understanding venom evolution more broadly. Sequencing the transcriptome of venom-producing glands has greatly increased the ability of researchers to examine and characterize the toxin repertoire of small taxa with low venom yields. Here, we use RNA-seq to characterize the Duvernoy’s gland transcriptome of the Plains Black-headed Snake, *Tantilla nigriceps*, a small, semi-fossorial colubrid that feeds on a variety of potentially dangerous arthropods including centipedes and spiders. We generated transcriptomes of six individuals from three localities in order to both characterize the toxin expression of this species for the first time, and to look for initial evidence of venom variation in the species. Three toxin families—three-finger neurotoxins (3FTxs), cysteine-rich secretory proteins (CRISPs), and snake venom metalloproteinases (SVMPIIIs)—dominated the transcriptome of *T. nigriceps*; 3FTx themselves were the dominant toxin family in most individuals, accounting for as much as 86.4% of an individual’s toxin expression. Variation in toxin expression between individuals was also noted, with two specimens exhibiting higher relative expression of c-type lectins than any other sample (8.7–11.9% compared to <1%), and another expressed CRISPs higher than any other toxin. This study provides the first Duvernoy’s gland transcriptomes of any species of *Tantilla*, and one of the few transcriptomic studies of RFS not predicated on a single individual. This initial characterization demonstrates the need for further study of toxin expression variation in this species, as well as the need for further exploration of small RFS venoms.

## 1. Introduction

Venom has independently evolved over 100 times across the Tree of Life, amounting to more than 200,000 species which use this protein and peptide mixture for prey capture and predator defense [1,2,3,4]. Snakes have evolved several different venom delivery systems primarily for prey capture [5,6]. Solenoglyphous and proteroglyphous (“front-fanged”) taxa—those with moveable or fixed hollow front fangs that deliver venom from a pressurized venom gland, such as vipers and elapids—have been focal systems in venom research, providing invaluable insight into ecology and natural history [7,8,9,10,11], evolution and phylogenetic patterns [12,13,14,15], behavior and physiology [16,17,18], and importantly the global snakebite epidemic [19,20]. Studies expanded to include distinct populations and range-wide sampling have further provided insight into intraspecific venom variation and gene evolution [13,21,22,23,24]. Not surprisingly, this focus on charismatic and medically-significant vipers and elapids has greatly outpaced the body of literature on the venoms of opisthoglyphous (“rear-fanged”) colubrid snakes, the majority of which are not considered dangerous to humans [25,26].

Many rear-fanged colubrid snakes (RFS) are capable of immobilizing and incapacitating prey by means of venom produced in a Duvernoy’s gland (DVG) posterior to the eye [12,27,28,29,30,31,32]. Typically, these branched serous glands have neither an associated muscle insertion (for pressurization) nor the capacity for storage of large amounts of its toxic secretion [29,32]; instead, a duct system funnels these low-pressure venoms to solid, grooved rear fangs to then be delivered into captured prey [29,32,33]. RFS are increasingly being recognized as important pieces largely missing from the broader context of venom evolution, variation, and ecological significance [25,26,31,34,35,36,37]. When modern transcriptomic and/or proteomic methods have been applied, RFS have been shown to harbor tremendous toxin diversity, including novel phenotypes (e.g., [38]), prey-specific toxins (e.g., [39,40]), and novel components not previously considered venoms (e.g., [41]).

Historically, the small size of some RFS species and low venom yields from their DVG have limited the ability of researchers to characterize their venom phenotypes and genotypes, leading to the substantial gap in knowledge of venom composition between RFS and front-fanged taxa. Modern transcriptomic methods can overcome the limitations of small input size [42], and have been utilized to effectively characterize venoms from miniscule taxa, including a number of venomous invertebrates (e.g., [43,44,45]). Still, transcriptomic analyses of snake venom glands have been almost entirely limited to larger-bodied species and/or those whose venom is considered medically-significant to humans ([46] and references therein). To this point, a recent summary of snake venom gland transcriptomes by Modahl et al. [46] noted that RFS account for only 23% of species with published transcriptomes (15 of 65), despite accounting for approximately two-thirds of snake biodiversity [47,48]; almost all of these RFS with sequenced glands are larger bodied taxa. Thus, a substantial percentage of venom-producing snakes have not been studied with modern methods, but are crucial to our understanding of venom evolution, function, and diversity in need of further exploration.

Snakes in the genus *Tantilla* are one such clade of RFS that are known to produce venom, but the composition of their venom remains almost completely unknown [27,49,50,51,52,53]. The genus includes 67 species of small-bodied (most <25 cm total length), semi-fossorial, opisthoglyphous colubrids distributed across North and Central America [48,54,55]. Although natural history data is lacking for many species due to their cryptic nature, *Tantilla* are broadly considered to primarily consume arthropods, including potentially dangerous prey items such as centipedes, scorpions, and spiders [51,56,57,58]. *Tantilla* possess a DVG, grooved and enlarged rear fangs, and a low-pressure venom yield, allowing them to subdue prey items as quickly as or faster than venomous front-fanged species take down prey [28,29,56,59,60]. Their small size, reluctance to bite, and very low venom yield precludes these snakes from posing any threat to humans, and makes them challenging to study [49,52].

*Tantilla* venom has largely gone unstudied; only two species—*T. nigriceps* and *T. cucullata*—have been investigated in any detail as part of broader proteomic studies [40,52,53]. *Tantilla nigriceps* is widely distributed in the Chihuahuan Desert and Great Plains of the central United States and northern Mexico [56,61] and are known to predate potentially dangerous prey items including spiders, centipedes, scorpions, millipedes, and various insects [57,62]. Using enzymatic assays, Hill and Mackessy [53] detected low levels of snake venom metalloproteinase (SVMP) activity in two samples of *T. nigriceps* venom as part of a larger study across North American colubrids (see also Saviola et al. [63]). Additionally, Hill and Mackessy [53] recovered a fragment of a peptide hypothesized to be a novel vascular endothelial growth factor toxin (VEGF). An SDS-PAGE profile including a *T. cucullata* venom sample was published as a supplemental figure in Modahl et al. [40], providing evidence of SVMPIIIs and CRISPs in the venom of that species. Beyond these initial works, no further attempts to fully characterize any *Tantilla* venom have been published, and more generally, no DVG transcriptomes have been sequenced from any species of this genus.

Given this void in our understanding of small RFS venoms, we chose to characterize the Duvernoy’s gland transcriptome of the Plains Black-headed Snake, *Tantilla nigriceps*, to provide new insight into the toxin composition of these snakes. To this end we utilized mRNA sequencing to characterize the DVG transcriptomes from six individuals across three localities, and tested for differences in expression across these locations. Based on the only previous study of *T. nigriceps* venom, we predicted SVMP transcripts would dominate toxin expression in DVG transcriptomes, with additional toxin families such as cysteine-rich secretory proteins (CRISPs) being present, but more lowly expressed. We also predicted we would find subtle intraspecific variation in venom expression between samples from different localities, but not significant differential expression between samples, as the natural history of the species (as currently understood) is similar across the areas sampled.

## 2. Results

We sequenced the DVG transcriptomes of six adult *Tantilla nigriceps* collected in Texas and New Mexico (Table 1; Figure 1) using 150 base-pair (bp) paired-end transcriptome sequencing on the Illumina NovaSeq and NextSeq platforms. Summary statistics for the six DVG transcriptomes are in Table 1. For each individual, we generated 8,088,121–28,872,668 read pairs (average: 20,075,935 ± 7,332,393) of which 73.7–83.4% were merged (average: 79.6 ± 3.1%). Following assembly, annotation, duplicate and chimera removal, and clustering, we combined the annotated transcripts from each assembly into a consensus transcriptome. This consensus transcriptome consisted of 36 unique putative toxin transcripts–primarily comprising four toxin families–and 2409 unique nontoxin transcripts (Table 2; Figure 2). The unique toxins accounted for 63.7% of the consensus expression based on RSEM-mapped reads across the six individuals (Appendix A).

The *Tantilla nigriceps* DVG transcriptome is dominated by three toxin families: three-finger toxins (3FTxs), cysteine-rich secretory proteins (CRISPs), and snake venom metalloproteinases (SVMPIIIs) (Table 2). Two individuals exhibited relatively high expression of c-type lectins (CTLs), as well. 3FTxs were especially abundant, with nine unique transcripts accounting for more than 33% of the average total transcriptome expression and more than 54% of average toxin expression. Two unique CRISP transcripts accounted for a combined 14.6% of average total transcriptome expression and 24.0% of toxin expression, whereas the eight unique SVMPIIIs collectively accounted for 11.2% of total transcriptome expression and 17.5% of toxin expression. The seven unique CTLs recovered accounted for 1.9% of total transcriptome expression and 3.1% of toxin expression. Eight additional toxin families—some of which may have no function or only indirect toxin function [65]—were recovered in much lower abundance (<1% of both total and toxin expression): three acetylcholinesterases (AChEs), one fused toxin, one Kunitz-type proteinase inhibitor (KUN), one phosphodiesterase (PDE4), one phospholipase-A2 (PLA2), one phosopholipase-B (PLB), one vascular endothelial growth factor (VEGF), and one waprin.

Broadly, the five individuals collected in the Chihuahuan Desert expressed similar toxin transcriptomes dominated by the aforementioned 3FTxs (34.9–72.8% of toxin expression), CRISPs (11.0–39.8%), and SVMPIIIs (7.1–27.9%). Conversely, the one individual collected in the Great Plains—ASNHC 15180—expressed the highest proportion of 3FTxs (86.4%), as well as a substantially higher proportion of CTLs (11% of toxin expression), with less than 1% of the toxin expression dedicated to CRISPs or SVMPIIIs (Figure 1; Table 2; Appendix A). CTLs accounted for more than 1% of the total toxin expression in only one individual from the Chihuahuan Desert (ASNHC 15179: 8.7%).

We tested for significant differences in toxin expression across life history traits using DESeq2 [66] and edgeR [67] as a preliminary look into intraspecific venom variation. No significant differences between sex or state of origin, or across SVL, were detected concordantly by both analyses (Appendix A). Testing latitude as a continuous variable, 13 toxin transcripts (six 3FTxs, both CRISPs, and five SVMPIIIs) and two toxin families (CRISPs and SVMPIIIs) were considered significantly differentially expressed; across longitude, two transcripts (a 3FTx and the VEGF) and one family (VEGF) were significantly differentially expressed (Appendix A). Clearly, more complete sampling across the distribution of the species is needed to determine if any of these significant differences (or the lack thereof) are indicative of broader patterns of venom variation in *T. nigriceps*, or artifacts of sampling bias.

## 3. Discussion

Our study provided the first look into the DVG transcriptome of any species of *Tantilla*, and a major step towards the characterization of *T. nigriceps* venom. In contrast to our predictions, *T. nigriceps* venom expression was not dominated by SVMPs, but instead by short neurotoxic 3FTxs in most individuals. Indeed, nine 3FTx transcripts accounted for more than half of the average toxin expression of *T. nigriceps* (Table 2) and more than a third of the overall expression in the DVG. CRISPs and SVMPIIIs were also highly expressed, but not to the level of 3FTxs in most individuals.

RFS are generally characterized as having either an elapid-like neurotoxic venom (dominated by 3FTxs), or an enzymatic, viper-like venom (dominated by SVMPs) [26,68]. For example, sixteen unique 3FTxs together make up 60% of the elapid-like transcriptome of *Spilotes sulfureus*; an SVMP is present, but accounts for only 2% of the total toxin expression. Conversely, SVMPs dominate the transcriptomes of *Ahaetulla prasina* (62%) and *Borikenophis portoricensis* (70%; [69], as well as *Hypsiglena sp.* (68%; [68]); 3FTxs, while present, are more lowly expressed (17%, 1%, and <1% respectively). As 3FTxs were the dominant toxin family recovered in the consensus transcriptome–as well as in five of the six individuals sequenced–our results suggest that the venom of *Tantilla nigriceps* is most similar to the elapid-like, 3FTx-dominant category. However, the venom does not necessarily fall cleanly into this category given the individual variation recovered. 3FTxs accounted for 34.9–86.4% of the individual toxin transcriptomes (Table 2). CRISPs and SVMPIIIs were highly variable in expression, and in one individual CRISPs were the most highly expressed toxin family. This diversity of highly expressed SVMPIIIs, 3FTxs, and CRISPs is reminiscent of *Boiga irregularis*, whose transcriptome is dominated by these same three toxin families, each with transcripts contributing substantially to the overall transcriptome [68].

Capturing variation in venom expression in *T. nigriceps* was only possible by having multiple samples from different localities. Few studies have generated RFS transcriptomes from more than a single individual, thus precluding any look at intraspecific variation in expression. Differences in toxin expression were evident between our samples from different localities (Figure 1). The lone sample from Duval County, Texas, for example, largely lacked SVMPs and CRISPs, instead expressing CTLs as its second-most dominant toxin family (Table 2) and expressing 3FTxs as a higher proportion of its transcriptome than any other sample. Moderate expression of SVMPs in other samples—especially in those from the Chihuahuan Desert—demonstrate the necessity of more complete transcriptomic sampling across its range. Transcriptomes from populations to the north (e.g., Colorado, Nebraska, and Kansas) and south (Mexico) would paint a clearer picture of venom expression variation as it relates to biogeographical (and potentially ecological) differences. Expression differences across individuals and habitats could be local adaptations in response to environmental or ecological selective pressures, or simply intraspecific variation that is difficult to explore with limited sample sizes. Though the difficulty of venom extraction and low yields from small RFS hinders proteomic characterization of *T. nigriceps* venom, these methods (undertaken by patient researchers) are critical to uncovering the functional activity of venom of these snakes.

By sequencing multiple individuals, we also hoped to recover an exact transcript match to the 3.5 kD peptide fragment discovered by Hill and Mackessy [53]. They hypothesized it to be a novel VEGF toxin based on the alignment of six of the 14 residues to a human VEGF sequence but were unable to otherwise identify it conclusively. We were unable to find an exact match to the fragment in our consensus transcriptome, but did align the same six residues with a VEGF transcript recovered in our transcriptome. This VEGF was lowly expressed (<0.0001% of average toxin TPM) and may not be a major component of the venom phenotype. Interestingly, we were able to align seven residues with transcript 3FTx-5 and six with 3FTx-1—the fifth- and first-most highly expressed 3FTxs based on average TPM. As 3FTxs were highly expressed in all individuals (and thus likely a major component of the venom phenotype), Hill and Mackessy [53]’s fragment might instead be a portion of a short neurotoxin. Characterization of transcriptomes from additional localities might clarify the identity of this toxin fragment and its role in the venom of *T. nigriceps*.

It is clear that further exploration of *Tantilla nigriceps* venom—and the venom of other species of *Tantilla*—is warranted. Variation in the venom transcriptome and venom proteome across the genus is likely given their phylogenetic and ecological diversity. To this point, Modahl et al. [40]’s SDS-PAGE profile for *T. cucullata* did not provide any evidence of 3FTxs, the most abundant toxin family in the transcriptome of *T. nigriceps*. While *Tantilla* are broadly considered to predate predominantly centipedes [58,70], some species are known to consume primarily beetle larvae [71] or a wider variety of arthropod prey including spiders, scorpions, and potentially even other small snakes [62]. Differences in the danger posed by certain prey items (e.g., centipedes) relative to others (e.g., beetle larvae) might correlate to compositional and/or functional variation of the venoms of different taxa, or different populations of the same taxon.

Small rear-fanged snake species represent an informative but neglected component in our understanding of venom evolution. Although this gap in knowledge has been pointed out numerous times (e.g., [25,26,31,34,37]), the vast majority of small RFS venoms still have not been afforded even cursory investigations. This dichotomy between recognized importance and continued neglect is underscored by the fact that, when studied, colubrid venoms are a source of incredible toxin diversity and even novel toxins (and toxin families). Transcriptomic and proteomic studies of colubrid venoms have also unlocked important new insights into the ecology, evolution, and natural history of the snakes themselves (e.g., [38,39,40,69]). Without an understanding of these key, overlooked components, the broader picture of venom evolution as a whole will remain incomplete.

## 4. Conclusions

In this study, we present the first Duvernoy’s gland transcriptomes of six *Tantilla nigriceps* from three localities in the southern United States. Three-finger toxins, cysteine-rich secretory proteins, and snake venom metalloproteinases were the most highly expressed toxin families, and two snakes also exhibited moderate expression of c-type lectins. Our results provide the first transcriptomic characterization of the Duvernoy’s gland of any *Tantilla* species, and highlight the need for further investigation into the venom of these and other small, semi-fossorial rear-fanged snakes.

## 5. Materials and Methods

### 5.1. Sample and Gland Collection

Six adult *Tantilla nigriceps* were opportunistically collected from Texas in June 2017 and New Mexico in August 2018 (Figure 1; Table 1). Five individuals were collected from the Chihuahuan Desert and one was collected from the Great Plains. We collected venom following a modified protocol similar to Rosenberg [33] and Hill and Mackessy [52]. Briefly, snakes were anesthetized using isoflurane until fully relaxed, and a 6 μg/g dose of pilocarpine was subcutaneously injected into the dorsolateral aspect of the snake, approximately 2 cm posterior to the head. Saliva from the oral cavity was collected using a micropipette, and venom was collected by using thin capillary tubes and/or 2–20 μL micropipette tips placed over the enlarged rear fangs. Due to the small size of the snakes—none weighed more than 2.5 g—very low venom yields were recovered. Four days after venom extraction, snakes were euthanized using a two-step MS-222 injection [72]. Both the left and right DVG of each individual were excised and immediately placed in RNAlater, briefly stored at 4 °C, and then transferred to long-term storage at −80 °C. Snakes were collected and research carried out under the following permits and protocols: New Mexico Department of Game and Fish Scientific Collecting Permit (NM SCP# 3697), Texas Parks and Wildlife Scientific Permit for Research (TX SCP# SPR-0390-029). Specimens were deposited in the Angelo State Natural History Collections (ASNHC). A CT scan of a *T. nigriceps* (UMMZ:Herps:69019) skull was retrieved from MorphoSource (ark:/87602/m4/M39216) [64].

### 5.2. RNA Extraction and Sequencing

We isolated RNA from the excised Duvernoy’s glands by means of a standard TRIzol extraction, as described by [73,74,75]. Briefly, DVG were finely minced, placed in TRIzol solution (Invitrogen), then homogenized and transferred to phase lock heavy gel tubes (5Prime). Total RNA was subsequently isolated using chloroform, then purified using isopropyl alcohol and ethanol precipitation. RNA was quantified using either a Qubit RNA BroadRange or High Sensitivity Kit, depending on the whether a quantifiable amount was detected on a first attempt. We used a Bioanalyzer 2100 and RNA 6000 Pico Kit (Agilent Technologies) to determine RNA quality and to ensure there was sufficient RNA to continue with cDNA library preparation and sequencing. From the isolated mRNA, we produced cDNA libraries using magnetic bead isolation of mRNA, and subsequent cDNA synthesis and PCR amplification. We used the NEBNext Poly(A) mRNA Magnetic Isolation Module (NEDB #E7490S) with equal input amounts of extracted RNA from the left and right glands of each individual to isolate mRNA. Following bead isolation and cleanup, we prepared cDNA libraries using a NEB Next Ultra RNA Library Prep Kit for Illumina (NEB #E7530). To achieve a target mean fragment size of 400 bp, we used a fragmentation time of 13 min, 30 s. For amplification of double-stranded cDNA libraries, we used 14 PCR cycles. We used a Qubit dsDNA High Sensitivity Kit and a Bioanalyzer 2100 with a DNA High Sensitivity Kit to determine library yield and quality. We determined the total concentration of amplifiable cDNA in each library using a KAPA qPCR (Roche KK4873). We performed a final concentration and quality check on the pooled libraries on the Bioanalyzer 2100 and via a KAPA qPCR. Pooled libraries were sequenced with 150 bp paired-end reads on either an Illumina NovaSeq 6000 platform at Florida State University College of Medicine Translational Science Laboratory (ASNHC 15178–15182) or an Illumina NextSeq 550 platform at the Clemson University Genomics and Bioinformatics Facility (ASNHC 15183).

### 5.3. Transcriptome Assembly and Annotation

We trimmed reads of base calls < 5 using Trim Galore! v. 0.4.4 (https://github.com/FelixKrueger/TrimGalore, accessed on 6 May 2019) and merged them using PEAR v. 0.9.10 [76]. We then performed de novo assembly using three different assembly methods, following the recommendations of Holding et al. [77]: Extender [73], SeqMan NGen v. 14 (using the Lasergene DNAStar software package; Madison, WI, USA: https://www.dnastar.com/t-nextgen-seqman-ngen.aspx, accessed on 6 May 2019), and Trinity v. 2.0.3 [78]. This combination of assemblers best captures the full repertoire of toxin and nontoxin genes from venom gland transcriptomes [77].

We annotated the assembled contigs from each assembler via blastx searches against the UniProt animal venom proteins and toxins database (http://www.uniprot.org/program/Toxins, accessed on 6 May 2019), using a minimum e-value of $10^{-4}$. We annotated both toxins and nontoxins by using cd-hit-est [79] to cluster sequences to a custom database of previously-annotated snake toxins and nontoxins; sequences and their associated signal peptides were automatically annotated when match percentages were >80%. We then manually annotated the remaining toxin and nontoxin contigs by comparing sequences to blastx results, as in [22]. Once annotation was complete, we combined annotated transcripts from the three assemblers and removed duplicates. We then screened for chimeric sequences by using BWA-MEM [80] to align merged reads to the annotated transcripts, removing reads with mismatches such as gaps or nucleotide differences. Any transcript with no coverage at any base was removed automatically. Additional transcripts were manually checked if the difference in the average length of reads on either side of a given site was greater than 75%. We then clustered the remaining transcripts of each individual using cd-hit-est at a threshold of 98% in order to reduce the redundancy of single locus allelic variation. Finally, we produced a species consensus transcriptome by combining the de novo-assembled transcriptomes of all six individuals and clustering them using cd-hit-est at a threshold of 95%. Transcripts were translated into amino acid sequences using the Sequence Manipulation Suite [81].

### 5.4. Transcriptomic Analyses

Using RSEM [82] with the default Bowtie 2 alignment settings [83], we mapped merged reads from each individual to the consensus transcriptome to determine expected counts (EC) of reads mapping to transcripts and calculate the normalized metric of transcripts per million reads (TPM). Zero-values were imputed using the ‘cmultRepl’ function in the zCompositions R package [84]. In addition to these metrics for each toxin transcript, we summed TPM and EC across paralogous toxins from the same gene family to visualize and analyze variation across toxin families. We visualized toxin composition of individuals and the consensus using TPM.

To investigate venom variation in *Tantilla nigriceps*, we used DESeq2 [66] and edgeR [67] to test for differential expression across life history traits. We tested for differences across body size (SVL), sex, and location. Five of our samples were collected from the Chihuahuan Desert, while a single individual was from the Great Plains in southern Texas. Without biological replicates, we did not have the statistical power to test for differential expression between individuals in the different biomes, so we instead used latitude and longitude as continuous variables in order to see if, preliminarily, there were differences that may be ascribed to those areas worth further investigation in the future. Expected counts for each venom transcript and summed expected counts for toxin families were used for these analyses. For DESeq2 analyses, we performed Wald significance tests with a local fit of dispersions and corrected *p*-values for false-discovery rate (FDR). For edgeR analyses, we fit negative binomial generalized linear models for each gene and tested for significance with likelihood ratio tests; *p*-values were corrected for FDR as in DESeq2 analyses. We considered genes or gene families significantly differentially expressed if they were found to have FDR-corrected α < 0.05 by both analyses.

## Figures and Tables

**Figure 1 toxins-13-00336-f001:**
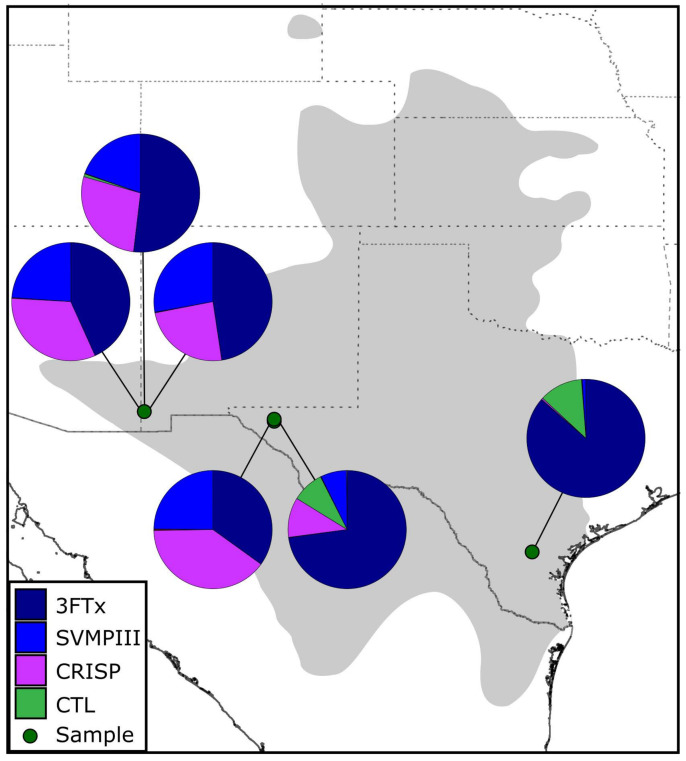
Sampling localities and associated DVG transcriptome of *Tantilla nigriceps* used in this study. The shaded gray area on the map represents the approximate range of *T. nigriceps* modified from Ernst and Ernst [56]. Pie charts indicate the proportional contribution of each of the four major toxin families recovered to the overall toxin transcriptome.

**Figure 2 toxins-13-00336-f002:**
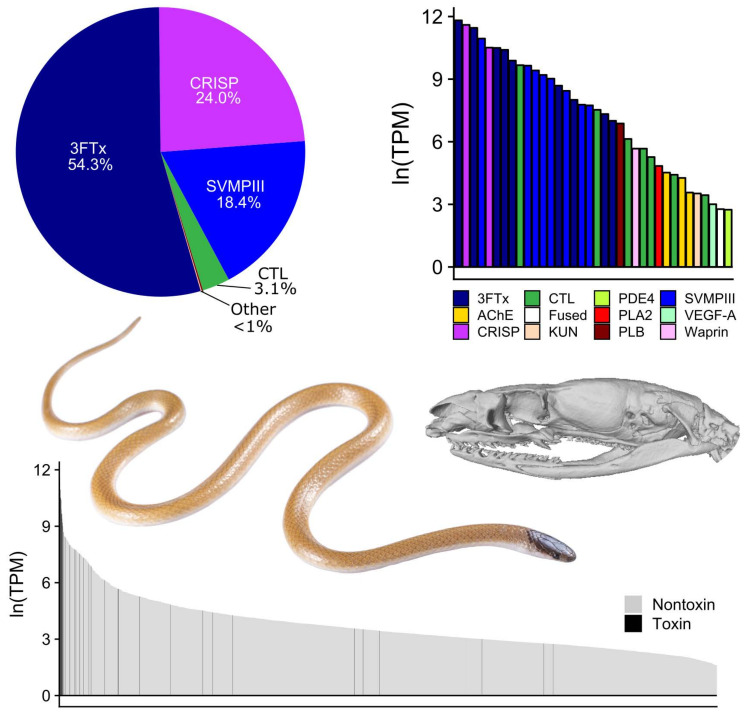
Expression of the Duvernoy’s gland transcriptome of *Tantilla nigriceps*. Data is the average of six individuals (see Appendix A for individual data). The pie chart represents proportional toxin gene expression by toxin family. The colored barchart represents expression of each toxin transcript, and the bottom black-and-white graph represents toxin and nontoxin gene expression in the Duvernoy’s gland. Photo of *T. nigriceps* in life by T. Schramer. Inset skull scan of UMMZ:Herps:69019 accessed via MorphoSource (ark:/87602/m4/M39216) [64]; skull not to scale of inset photo.

**Table 1 toxins-13-00336-t001:** Metadata and sequencing outputs of *Tantilla nigriceps* used in this study.

ASNHC No.	Field ID	Locality	Sex	Read Pairs	Merged Reads
15178	CLP2590	Hudspeth Co., TX, USA	F	27,832,797	22,093,170
15179	CLP2591	Hudspeth Co., TX, USA	M	28,872,668	24,078,211
15180	CLP2592	Duval Co., TX, USA	F	22,736,233	17,826,005
15181	CLP2753	Hidalgo Co., NM, USA	F	14,445,533	10,648,351
15182	CLP2754	Hidalgo Co., NM, USA	M	8,088,121	6,232,048
15183	CLP2759	Hidalgo Co., NM, USA	M	18,480,258	15,030,748

**Table 2 toxins-13-00336-t002:** Percent contribution of the most highly expressed toxin families to the overall toxin transcriptome. Percentages calculated by summing individual toxin transcripts by family and dividing by the total toxin TPM. Values rounded to the nearest 0.1%.

	ASNHC	ASNHC	ASNHC	ASNHC	ASNHC	ASNHC	Avg.
	15178	15179	15180	15181	15182	15183	
**3FTx**	34.9%	72.8%	86.4%	51.9%	47.6%	43.3%	54.3%
**CRISP**	39.8%	10.9%	0.4%	27.6%	24.3%	32.7%	24.0%
**SVMPIII**	24.8%	7.1%	1.0%	19.6%	27.9%	23.9%	18.4%
**CTL**	<0.1%	8.7%	11.9%	0.8%	<0.1%	<0.1%	3.1%
**Others**	0.4%	0.43%	0.3%	0.1%	0.2%	0.1%	0.2%

## Data Availability

The data underlying this article are available in its online supplementary material and the National Center for Biotechnology Information (NCBI). RNA sequencing data were submitted to the Sequence Read Archive (SRA) under BioProject (PRJNA88989). BioSamples accession numbers (SAMN18863717–SAMN18863722). SRR accession numbers (SRR14319402–SRR14319407).

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
