# Peer review of "Duvernoy’s Gland Transcriptomics of the Plains Black-Headed Snake, Tantilla nigriceps (Squamata, Colubridae): Unearthing the Venom of Small Rear-Fanged Snakes"

_toxins, 2021, doi:10.3390/toxins13050336_

Round 1

Reviewer 1 Report

I found the paper to be of very high quality in both its conception and realization, and I have no hesitation in recommending it for publication. The study is on a species of rear-fanged colubrid which is a family of venomous snakes have been practically ignored by most venom researchers. This paper therefore makes a valuable contribution to our knowledge. The study design used multiple individuals, so a high degree of confidence can be attributed to the findings. Also intra-specific variation was investigated adding value to the paper. The only interpretation I would challenge (and this is a minor criticism - the only one I could find), is line 170-171 (Our results suggest that the venom of Tantilla nigriceps does not fall cleanly into one of these two categories).

I would argue (as mentioned by the authors), that colubrid (RFS) venoms are either SVMP dominant e.g. Dispholidus/Ahaetulla/ Borikenophis/ Thamnodynastes, or 3FTx dominant - e.g. Rhinobothryum/ Trimorphodon/ Boiga.

Therefore at 54.3% 3FTx, Tantilla would be classed as 3FTx dominant wouldn't it? (albeit also expressing high levels of CRiSP -24%).

Previous publications have shown CRiSP to be the 3rd most highly expressed protein family in colubrid venoms (mean 5%, maximum 19%, averaged from 9 species), So Tantilla nigriceps venom is unusual in its high expression of that protein family - so an interesting finding if the transcriptome expression matches the venom gland proteome expression for this protein family. When the venoms of more species of RFS are characterized, I would expect to see more species expressing high levels of CRiSP.

The transcriptomics results are well presented and easy understandable, and all findings were clearly expressed and easy to follow.

Happy to recommend for publication.

Author Response

We modified the sentences as suggested.

Reviewer 2 Report

This manuscripts reports the generation and detailed characterization of the Duvernoy’s gland transcriptome from Tantilla nigriceps, a small rear-fanged venomous snake. They stress the value of using transcriptomic studies to characterize and study venoms from small and low-yield animals, and include characterization and comparison of six animals from three localities, providing insight into intraspecific variation in venom phenotypes. The latter is a particularly nice aspect of the study, as there is indeed variation between several individuals - had only a single of these samples been characterized, it may have resulted in a skewed and incomplete understanding of venom composition in this species. Overall, I think this is a well-designed study and well-written manuscript that will be of interest to Toxins readership, and find it suitable for publication. I have no major concerns or suggestions for the authors.  

Author Response

no comments